# Procedural Tools and Technics for Transcatheter Paravalvular Leak Closure: Lessons from a Decade of Experience

**DOI:** 10.3390/jcm12010119

**Published:** 2022-12-23

**Authors:** Sébastien Hascoët, Grzegorz Smolka, Teoman Kilic, Reda Ibrahim, Eustaquio-Maria Onorato, Patrick A. Calvert, Didier Champagnac, Xavier Freixa-Rofastes, Aleksejus Zorinas, Juan Pablo Sandoval, Gregory Ducrocq, Frederic Bouisset, Alain Fraisse, Benoit Gerardin

**Affiliations:** 1Marie Lannelongue Hospital, Groupe Hospitalier Paris Saint Joseph, Faculté de Médecine Paris-Saclay, Université Paris-Saclay, BME Lab, 133 Avenue de la Résistance, 92350 Le Plessis Robinson, France; 2Royal Brompton Hospital, Sydney Street, London SW3 6PY, UK; 3Department of cardiology, Medical University of Silesia—Poniatowskiego 15, 40-055 Katowice, Poland; 4Department of Cardiology, Medical Faculty, Kocaeli University, Umuttepe, Yerteskesi, Kocaeli 41380, Turkey; 5Department of Cardiology, Montreal Heart Institute, Université de Montréal, 5000 Belanger Street, Montreal, QC H1T 1C8, Canada; 6Monzino Cardiological Center, IRCCS, 20138 Milan, Italy; 7Department of Cardiology, Royal Papworth Hospital, NHS Foundation Trust, University of Cambridge, Papworth Road, Trumpington, Cambridge CB2 0AY, UK; 8Médipôle Lyon Villeurbanne, 158 rue Léon Blum, 69100 Villeurbanne, France; 9Interventional Cardiology Department, Hospital Clinic of Barcelona, University of Barcelona, 08306 Barcelona, Spain; 10Vilnius University Hospital Santaros Klinikos, Vilnius University, 08410 Vilnius, Lithuania; 11Ignacio Chávez National Institute of Cardiology, Universidad La Salle, Mexico City 14080, Mexico; 12Bichat Hospital, Assistance Publique des Hôpitaux de Paris, Hôpital Bichat-Paris, 46 rue Henri Huchard, 75018 Paris, France; 13Department of Cardiology, Toulouse Rangueil University Hospital, UMR 1295 INSERM, Hôpital Rangueil, CHU Toulouse, 1 Avenue du Pr Jean Poulhès, 31000 Toulouse, France

**Keywords:** paravalvular leak, mitral valve, aortic valve, catheterization, plug

## Abstract

Prosthetic paravalvular leaks (PVLs) are associated with congestive heart failure and hemolysis. Surgical PVL closure carries high risks. Transcatheter implantation of occluding devices in PVL is a lower risk but challenging procedure. Of the available devices, only two have been specifically approved in Europe for transcatheter PVL closure (tPVLc): the Occlutech^®^ Paravalvular Leak Device (PLD) and Amplatzer™ ParaValvular Plug 3 (AVP 3). Here, we review the various tools and devices used for tPVLc, based on three observational registries including 748 tPVLc procedures performed in 2005–2021 at 33 centres in 11 countries. In this case, 12 registry investigators with over 20 tPVLc procedures each described their practical tips and tricks regarding imaging, approaches, delivery systems, and devices. They considered three-dimensional echocardiography to be the cornerstone of PVL assessment and procedure guidance. Anterograde trans-septal mitral valve and retrograde aortic approaches were used in most centres, although some investigators preferred the transapical approach. Hydrophilic-coated low-profile sheaths were used most often for device deployment. The AVP 3 and PLD devices were chosen for 89.0% of procedures. Further advances in design and materials are awaited. These complex procedures require considerable expertise, and experience accumulated over a decade has no doubt contributed to improve practices.

## 1. Introduction

Paravalvular leaks (PVLs) occur around surgically or percutaneously implanted valves, in 6% to 32% of cases [1,2,3]. PVLs are more common at the mitral than the aortic valve and with mechanical than biological valves [4]. Symptomatic heart failure or mechanical haemolytic anaemia develops in 1% to 5% of patients with PVL [5]. The standard treatment for these complications was open surgery, which involved considerable risks including up to 10% mortality [6,7,8]. Percutaneous paravalvular leak closure (tPVLc), first performed in 1992 [9] has emerged as an attractive alternative [10,11,12]. The development of many implantable occluders, combined with advances in imaging and the accumulation of experience, have contributed to foster progress in tPVLc [13,14].

However, tPVLc is a complex and demanding procedure, [15,16,17] more so at the mitral than the aortic valve. Technical difficulties may occur in crossing the leak with the wire and delivery sheath, obtaining stable implantation, and ensuring that the device provides complete occlusion without impairing valve motion. Only two devices have been specifically approved for tPVLc, the Paravalvular Leak Device (PLD) (Occlutech GmbH, Jena, Germany) and the Amplatzer ParaValvular Plug 3 (AVP 3) (Abbott Medical, Plymouth, MN, USA). Nonetheless, many other devices such as vascular plugs and, to a lesser extent, congenital-heart-defect occluders are used also. Some delivery sheaths recommended by manufacturers have a suboptimal profile and insufficient push ability and flexibility for this exacting procedure. The accumulation of experience over time has allowed interventional cardiologists to develop ingenious stratagems for overcoming technical challenges.

The objective of this article was to describe practical tips and tricks for tPVLc based on experience acquired in Europe and North America (1 centre).

## 2. Methods

We identified investigators with extensive experience in tPVLc by searching three observational registries including 748 procedures carried out between 2005 and 2021, at 33 centres in 11 countries. The retrospective observational FFPP registry included 386 procedures in 366 patients between 2005 and 2019 (NCT05117359) and the prospective FFPP registry 238 procedures in 216 patients between 2017 and 2019 (NCT05089136) [14]. The European Para Valvular Leak closure (EuroPVLc) registry started in 2020 is still recruiting (NCT05506293) and has recorded data for 124 patients until the end of 2021.

In this case, 15 operators who contributed patients to the registries were found to have performed more than 20 tPVLc procedures each and were invited to participate in the study. Among these experts, 12 accepted and completed a questionnaire before participating in an interview about procedural tips and tricks. These were divided into categories based on whether they pertained to pre-procedural imaging, procedure guidance, the approach, crossing the leak, the delivery systems, or the devices. When discordances were noted between expert reports and manufacturer information, we performed some bench tests to assess the relevant device-sheath couple.

## 3. Results

### 3.1. Pre-Procedural Imaging

#### 3.1.1. Echocardiography

Echocardiography, notably via the trans-oesophageal route (TEE), was identified as the key imaging technique for assessing PVL morphology and severity. All 12 experts relied on 2D and 3D TEE to assess PVL position, particularly for mitral valve, and plan the tPVLc strategy (Table 1).


*Tips and tricks*


Authors suggested that the same, specifically trained echocardiographer should perform the pre-procedural TEE and intra-procedural echocardiography guidance.

#### 3.1.2. Cardiac Computed Tomography (CT)

CT was not routinely used. It could provide a morphological assessment of the leak, but it was limited by valve artifacts. Considerable variability in the use of CT was noted among experts (Table 1), with routine CT and merging with fluoroscopy at one end of the spectrum and never using CT at the other. Some experts reported using CT more often for aortic tPVLc given the possible difficulties with TEE evaluation at this site, notably after trans-aortic valve implantation. CT was also useful in some cases to identify the PVL location and anticipate the best C-arm angulation on fluoroscopy upfront, particularly for tPVLc without TEE guidance.


*Tips and tricks*


Authors suggested that fusion imaging to overlay CT and fluoroscopy images may help in targeting small PVLs.

#### 3.1.3. Cardiac Magnetic Resonance Imaging (MRI)

Cardiac MRI was not carried out routinely, but its potential usefulness was under investigation. Each of the 12 experts reported using MRI only very rarely or not at all (Table 1). Some authors suggested that MRI, notably using the 4D-flow technique, could supply dynamic information on the leak, quantify the regurgitation, and enabled measurement of the effective orifice regurgitation area [13]. Valve artifacts and limited availability were the main drawbacks.

#### 3.1.4. 3D-Printing

In very few cases, 3D-printing of CT images was performed to obtain a simulator on which devices can be tested to select the closest match to the leak. However, this time consuming and costly technique was used only very rarely or not at all by the 12 experts. Several experts, however, suggested that it was a promising tool for the most complex cases [14,18].


*Tips and tricks*


Bench testing on 3D-printed models was carried out in few challenging cases to select the optimal device prior to the procedure [19].

### 3.2. Procedure for Percutaneous Paravalvular Leak Closure (tPVLc)

#### 3.2.1. Anaesthesia

General anaesthesia was usually performed when TEE guidance was required and a transapical approach intended. Several experts reported that aortic tPVLc was feasible under local anaesthesia, with fluoroscopy and transthoracic echocardiography guidance (Table 2). One expert, however, preferred to use the same protocol for all tPVLc procedures to avoid confusing the staff.

#### 3.2.2. Peri-Procedural Imaging Guidance

TEE was widely used to guide mitral tPVLc, almost always with 3D imaging. When using the transseptal approach, 3D-TEE improved the accuracy of the transseptal puncture then provided an en-face view of the mitral valve from the left atrium that was useful to assess PVL position and to guide its crossing. Importantly, special attention was taken to ensure that the guidewire passed through the PVL and not through the valve.

Intra-cardiac echocardiography was not used, notably because 3D modality was not available at this time.

Some experts reported occasionally using fluoroscopy alone for aortic tPVLc. Fusing echocardiography or CT with fluoroscopy was increasingly used [14,20,21,22,23,24,25].


*Tips and tricks*


Authors recommended that the interventional echocardiographer guiding the procedure must be skilled in PVL morphology assessment using 2D, 3D, and multiplanar reformatting images. For mitral tPVLc, another necessary skill was the rapid provision of a 3D, en-face view of the left atrium with anatomical orientation to assist navigation in the left atrium.

### 3.3. Paravalvular Leak (PVL) Approach

#### 3.3.1. Mitral Percutaneous Paravalvular Leak Closure (tPVLc)

Three main approaches were used for mitral tPVLc (Figure 1): the anterograde approach through a femoral vein and the inter-atrial septum, the retrograde approach through an artery, the aortic valve, and the left ventricle, and the transapical approach by puncture of the left ventricular apex. The wire-looping technique combine the anterograde and retrograde approaches by inserting the wire through one of the access vessels then snaring it with a lasso inserted through the other access vessel.


*Tips and tricks*


Authors enlighten that PVL localization was the key determinant to choose the primary approach for crossing and closing mitral PVL. The real-time 3D en-face view of the mitral valve was positioned in the surgical view with the aortic valve at the top of the mitral ring (12 o’clock) and the left atrial appendage (LAA) at approximately the 9 o’clock position.

The transseptal antegrade approach was usually preferred, particularly for a mitral PVL located anteriorly (near the aorta) or anterolaterally (in the proximity of left atrial appendage). In the presence of a medially or posteriorly located defect, a retrograde approach was considered given the sharp angulation observed when a transseptal approach is used. A low transseptal puncture enabled a successful PVL closure for medial mitral PVLs and remained the approach of choice for some operators. Alternatively, other authors used a transapical approach to overcome this issue and facilitate mitral PVL closure.

##### Anterograde Mitral Approach

The anterograde approach was considered the first-line strategy by most of the experts (Table 3). The site of the echo-guided transseptal puncture depended on PVL location, the aim being to obtain the most straightforward access to the leak. A posterior-inferior trans-septal puncture site was usually suitable for defects close to the left atrial appendage, whereas a posterior trans-septal site was considered when the defect was septal or posterior. Some experts started to use the Baylis VersaCross^®^ radio-frequency system (Baylis now Boston Scientific, Marlborough, MA, USA) to improve puncture accuracy or feasibility in thick septum. Steerable sheaths that can be positioned along the direction of the leak were used almost routinely. For crossing the leak, the mother-in-child technique was often used. However, other methods, alternative materials, and shortcuts were also available. The steerable sheath tip was aligned with the PVL on the 3D-TEE images. Next, using a multipurpose or Judkins right diagnostic catheter for orientation, a straight, hydrophilic-coated, 0.035-inch guidewire (GLIDEWIRE^®^, Terumo, Shibuya City, Tokyo, Japan) was advanced across the PVL. The catheter was advanced over the guidewire into the left ventricle, and the guidewire was then exchanged for a long, stiff guidewire. The catheter was retrieved. Finally, a delivery sheath or guiding catheter was advanced and used to implant the occluder.

##### Retrograde Mitral Approach

Although no longer often used, the retrograde approach was reported by some experts to be very helpful when targeting medial or posterior mitral PVLs (Table 3). The retrograde approach may be useful when the septum cannot be crossed safely (thickened, surgical patch, or occluder).

This approach is usually contraindicated when the aortic valve is mechanical [11,26]. However, careful insertion of a hydrophilic catheter and relatively thinner delivery systems may be carried out. The use of a hydrophilic coated distal-tip catheter to advance into the left ventricle through the central opening of the bileaflet mechanical valve was usually safe but potential hazards of the technique were related to retrograde crossing by large catheters of a monoleaflet mechanical valve [11].

A Judkins right coronary catheter, a modified pigtail catheter or an Amplatz left coronary catheter were used to cross the leak. The subsequent steps were similar to those for the anterograde approach.


*Tips and tricks*


A pre-shaped extra stiff wire usually provided a good support to advance the delivery system into the left atrium without creating an arteriovenous wire loop

A 110 cm-long sheath was often required for device implantation via this approach [27].

##### Arterio-Venous-Loop Mitral Approach

Arterio-venous looping was a complementary strategy to the anterograde or retrograde approach and was sometimes considered when additional support was required to successfully advance the delivery system across the mitral PVL. The hydrophilic-coated guidewire was passed through the PVL via the anterograde or retrograde approach. Its tip was then snared with a lasso inserted by the opposite approach and pulled back to create a rail through the mitral PVL. This provides strong support for advancing a delivery sheath, predominantly via the anterograde approach.


*Tips and tricks*


A smooth guidewire such as the Terumo GLIDEWIRE^®^ was recommended for forming the loop. To further prevent a razor effect of the wire on the valves, the operator can use “the kissing catheter technique”, in which a catheter is advanced opposite the delivery sheath until close contact is achieved. The position is maintained by two clamps at each end. The next step is synchronized delivery-sheath pushing and catheter pull-back across the leak.

##### Transapical Approach

The transapical approach was used only very rarely by most experts but was considered when the anterograde or retrograde approach failed. For a few experts, however, the transapical approach was the first-line strategy, particularly for large and/or septal mitral leaks (Table 3).

The transapical approach facilitated leak crossing and the deployment of large devices such as the PLD. It was seen as more invasive and carrying a higher risk of complications. However, a high level of operator experience and close teamwork between the heart surgeon and interventional cardiologist minimized the risk. The procedures were performed in a hybrid operating room. The left ventricle was exposed by the heart surgeon then punctured at the apex. A short sheath was inserted into the left ventricle. The leak was crossed with a wire and the sheath was then inserted into the left atrium. Short sheaths were sufficient.


*Tips and tricks*


One expert suggested that to reduce the risk of bleeding related to rib fracture and damage of the intercostal neurovascular bundle, atraumatic plastic soft tissue retractors should be used. Zorinas et al. recommended limiting use of a rigid Finochietto retractor only to pericardial adhesion dissection and its hitching to the skin. The rib spreader should then be removed [28].

The use of negative pressure wound therapy and adequate antimicrobial regimen reduced the risk of wound infection. Identifying the apex of the LV should be performed prior to transapical puncture to prevent iatrogenic damage to the apex of the right ventricle. The left coronary anterior artery should also be identified to avoid an unfortunate puncture. If multiple devices are to be implanted within the leak, two parallel transapical accesses could be created.

#### 3.3.2. Aortic Percutaneous Paravalvular Leak Closure (tPVLc)

Occluder implantation was considered easier to perform at the aortic than at the mitral valve, and technical success rates was higher. The retrograde approach via the femoral or radial artery was usually chosen (Table 4). The right radial artery access facilitated the crossing of defects located anteriorly and near the left coronary sinus and the femoral artery access the crossing of posterior defects. A Judkins right, multipurpose, or Amplatzer Left 1 catheter is inserted across the leak.

One expert used a surgically managed retrograde approach via the axillary artery in patients with marked atherosclerosis, calcification, and/or tortuosity of the aorta, body height greater than 180 cm, or a need for additional support or for a large device. In taller patients, alternatively, brachial arterial access was also considered in few patients. The transapical approach was also occasionally used for closure of both mitral and aortic PVLs during the same procedure.


*Tips and tricks*


A 110 cm-long sheath was often required to reach and cross the defect.

### 3.4. Guidewires

#### 3.4.1. Crossing the PVL

The hydrophilic-coated, 0.035-inch GLIDEWIRE^®^ (Terumo) was generally used to cross the PVL, and a straight tip was preferred. Alternatively, a 0.035-inch Roadrunner wire (COOK) is sometimes useful. A stiff wire was not recommended at this step. Angulated-tip guidewires and the application of adequate torque was considered when the PVL was tortuous. A J-shaped, hydrophilic-coated, 0.035-inch guidewire did not usually fit through PVLs. Coronary 0.014-inch guidewires were generally not used. Smolka and al. enlighten using these wires when very precise leak targeting was needed, to prevent unexpected tension and movement of the distal end of the catheter by a larger wire.

#### 3.4.2. Advancing the Delivery Catheter

Once the hydrophilic-coated, 0.035-inch guidewire was passed through the leak, since it did not provide sufficiently strong support for the delivery catheter, replacement by a stiffer guidewire was generally necessary, except with the transapical approach. The exchange usually consisted of advancing a 4-Fr or 5-Fr catheter through the leak on the hydrophilic-coated guidewire, which was then removed and replaced by a stiff 0.035-inch guidewire.


*Tips and tricks*


Careful attention was required to avoid injuring the left ventricular apex with the tip of the stiff guidewire. The tip should be 7 cm long and smooth. A super-stiff, 0.035-inch guidewire was manually pre-shaped at its distal end to create a single open curve. A pre-shaped stiff guidewire designed for trans-aortic valve implantation (typically a SAFARI^®^ from Boston Scientific (Marlbourough, MA, USA)or a CONFIDA^®^ from MEDTRONIC (Minneapolis, MN, USA) was an alternative to conventional, Amplatz Super-stiff, 0.035-inch guidewires that minimized the risk of ventricular injury, notably when the delivery sheath crossed the leak only with strong pushing manoeuvres that mobilized the guidewire in the ventricle. Advancing these stiff guidewires through the smooth 4-Fr catheter was sometimes tricky, due to the pre-shaped tip. Consequently, the 4-Fr catheter was frankly advanced through the leak, at the apex of the ventricle, to prevent it from jumping out of the ventricle when advancing the stiff guidewire. Preliminary experience was reported with the GLIDEWIRE® ADVANTAGE ™ guidewire that combined a smooth 25 cm distal portion featuring the original GLIDEWIRE® with hydrophilic coating and a stiffer nitinol core providing support to advance the delivery sheath, eliminating the need for multiple wire exchanges.

### 3.5. Delivery System

Delivery systems are chosen specifically for each procedure based on the profile of the occluding device, shape of the leak, and distance from the access site to the leak (Table 5).


*Tips and tricks*


When a delivery system was inserted inside a sheath (mother-in-child technique), it had to be longer than the mother sheath. With the transapical approach, a short introducer and delivery sheath were sufficient.

#### 3.5.1. TorqVue™/Trevisio Delivery Sheaths

TorqVue™ and more recently Trevisio were the standard delivery sheath used to deploy the Amplatzer devices available from Abbott (Plymouth, MN, USA) that were chiefly used for straightforward procedures.

#### 3.5.2. Destination™ (Terumo)

Destination™ (Shibuya-ku, Tokyo—Japan)introducers were often used given their favourable profile and good push ability. Their 90-cm maximal length limited their use, notably when the introducer was inserted into a steerable sheath.

#### 3.5.3. Flexor^®^

Flexor^®^ delivery sheaths from Cook Medical (Bloomington, IN) were very often used given their low profile, good flexibility, and an extensive length range up to 110 cm, facilitating mother-in-child technique.


*Tips and tricks*


Flexor^®^ sheaths with a rotating movable with a Tuohy-Borst Sidearm adapter (haemostatic Y valve) from COOK were used to facilitate the insertion of the device in the sheath.

#### 3.5.4. Guiding Catheter

Guiding catheters with the largest internal lumen diameters were used (Table 5) given their low profile and their 110-cm length.

#### 3.5.5. Steerable Sheath

Several types of steerable sheaths were used with various curves chosen based on left atrial volume [29]. When several devices were implanted on mitral PVL during the same procedure, larger steerable sheaths were used.


*Tips and tricks*


For the anterograde mitral approach, a large steerable sheath placed in the left atrium facilitated PVL targeting and provided additional support to facilitate delivery sheath crossing. Some steerable sheaths had to be advanced on 0.032-inch exchange guidewire. When a large sheath was used to insert simultaneously several delivery sheaths, caution had to be paid on the efficacy of the haemostatic valve to limit bleeding.

### 3.6. Devices

Table 6 and Figure 2 show the devices implanted in the patients recorded in the three registries, with the number for each device. In most recent registry, the AVP 3 and PLD accounted for 89.0% of implanted devices.

All the devices used for tPVLc shared similarities in design. All were self-expandable, made of braided nitinol mesh with shape memory. All were implanted using delivery catheters, being initially tethered by a cable allowing retrieval if the position was incorrect after deployment. The method for delivery was similar for all devices. A delivery sheath was positioned across the PVL distally in the heart cavity (left ventricle for mitral or aortic tPVLc, via the transseptal or retrograde approach, with the left atrium for mitral tPVLc via the transapical or retrograde approach, or aorta for aortic tPVLc via the transapical approach). The selected device was inserted through the sheath and advanced across the PVL within the sheath. Next, the distal part of the device was half deployed in the heart cavity, and the half-uncovered occluder and sheath were gently pulled back close to the leak. The next step was a tricky manoeuvre combining sheath pull-back with complete device deployment within the PVL, so that the occluder was anchored to the PVL channel and its surrounding structures. Positioning, stability, efficacy, and relationship of the device with the valve were assessed. If needed, the device could be recaptured by advancing the sheath then redeployed until optimally positioned. The device was then released from its delivery cable. The differences between the devices listed below pertain to the nitinol mesh and to device shape, size, and profile. The specific characteristics of each device are listed below in order of frequency of device use in the selected registries.

#### 3.6.1. Amplatzer ™ ParaValvular Plug 3 (Abbott Medical, Plymouth, MN, USA)

The AVP 3 accounted for 56.6% of implanted devices in the three registries. The oval shape of this self-expanding, triple-disc device minimized the risk of valve-leaflet impingement (Figure 2). The discs at each end are thin and have a diameter only slightly larger than that of the waist, which defines device size. The various values of the short and long axes of the oval waist produce nine different sizes (Table 7). The AVP 3 obtained the CE mark in January 2020 as a class III implant for mechanical PVL closure. A radio-opaque marker positioned laterally in the middle of the long waist facilitated guidance of device deployment. The limited maximal size did not produce sufficient stability and efficacy to close very large defects.


*Tips and tricks*


All AVP 3 devices were implanted through sheaths smaller than initially recommended by the manufacturer (Table 7).

In order to orientate the AVP 3 occluder in the expected position, only a clockwise rotation of the delivery system was carried out to prevent any unexpected unscrewing of the device during the manoeuvre.

#### 3.6.2. Paravalvular Leak Device (PLD, Occlutech GmbH, Jena, Germany)

The PLD accounted for 14.2% of implanted devices in the three registries. This device was specifically developed for tPVLc. There are two versions, one with two rectangular discs and an ellipsoid waist and the other with two square discs and a circular waist (Figure 2). Each disc contains a polyethylene terephthalate (PET) patch. The two discs are linked by a nitinol connection in either a waist or a twist configuration. The twist design provided greater conformability to defect shape, while the waist design was more suitable for large defects. When a second device was required, the twist design was generally chosen. The twist design was also preferred for small defects. The 35% smaller surface areas of both the rectangular and square versions compared to a round design minimized the risk of valve-leaflet impingement. Two gold radiopaque markers, one in each disc, improved fluoroscopic visibility, thereby facilitating accurate deployment across the defect. Combinations of the different shapes, connections, and sizes produce four types of devices, with 19 devices in all, allowing occlusion of nearly all PVLs, including large defects (Table 8). The rectangular version with a waist design was the most widely used PLD. The PLDs obtained European CE mark approval in 2014 but do not yet have FDA Premarket Approval [30,31,32,33,34]. Compared to AVP 3 devices of similar size, PLDs have a larger profile and required a larger delivery sheath. For these reasons, the transapical approach was usually recommended when targeting a large defect.


*Tips and tricks*


Achieving proper PLD devices’ orientation was sometimes challenging given the ball connection between the delivery cable and the device that did not allow for the rotation of the occluder. Keeping the device half-opened and rotate it with the delivery sheath was carried out to overcome this issue.

#### 3.6.3. Ventricular Septal Defect (VSD) Occluders

The Amplatzer Muscular VSD Occluder contributed 7.9% of implanted devices in the three registries. This round, self-expanding device has two discs of the same diameter linked by a thick waist that is smaller in diameter than the discs and includes a polyester patch. Seven devices defined by different waist diameters are available (Table 7). These devices are occasionally used for large defects, as an alternative to the PLD, but have not been approved by regulatory authorities in this indication. Furthermore, the device is quite rigid and, in the event of a residual leak, worsening haemolysis was reported. The use of VSD occluders declined. Some experts still used them occasionally, whereas others advocated against them for tPVLc.

#### 3.6.4. Amplatzer Vascular Plug (AVP) 2

The AVP 2 accounted for 7.7% of implanted devices in the three registries. This round, self-expanding, triple-disc device. The three discs have the same diameter, the two discs at each end being thin and the central disc thick. In this case, 11 diameters are available (Table 7). The AVP 2 was used occasionally for long tubular PVLs and off-label in countries where the AVP 3 are not commercially available.

#### 3.6.5. Amplatzer Septal Occluder (ASO)

The ASO contributed 5.7% of implanted devices in the three registries. In this round, self-expanding device, two discs are connected by a short waist that is smaller than the disc diameter and includes a polyester patch. (Table 7). These occluders were used before specific tPVLc devices were developed and do not have regulatory approval for tPVLc. The large disc diameters carry a risk of valve-leaflet impingement. Worsening haemolysis has been reported in patients with residual leakage. The ASO was rarely used for tPVLc in the most recent registry. Most experts discouraged the use of this device.

#### 3.6.6. Amplatzer Vascular Plug 4 (AVP4)

The AVP4 accounted for 3.4% of implanted devices in the three registries. This self-expanding device has two heart-shaped components whose wide aspects are connected to each other by a narrow waist. Five models with maximal diameters of 4 to 8 mm were available. All models were delivered through a 4-Fr braided diagnostic catheter with a 0.038-inch inner lumen (Tempo^®^, Cordis, Hialeah, FL, USA). The AVP4 was used to close small PVLs and residual leaks along a larger device when only a 4-Fr catheter can cross the defect (Table 9). The 8-mm size was generally used.

#### 3.6.7. How to Choose a Device

The devices recommended by the experts were consistent with those most often reported in the registries (Table 10, Figure 2). The AVP 3 and PLD, which have regulatory approval for tPVLc, were the most widely recommended by the experts. The AVP 2 was considered an option for aortic tPVLc. Selection of device size was based on measurements of the PVL waist and circumferential extension. These measurements were carried out at the PVL vena contracta visualized by colour flow Doppler.

The experts recommended care in not oversizing the PLD. Some of them slightly undersized this device to decrease the risk of device deformation and valve-leaflet impingement. With the rectangular version, the waist of the device should match the waist of the defect. If the leak channel is longer than 5 mm, the model with the waist design may become distorted, and the twist design was consequently preferable, given its flexible connection. The PLD is large, and multiple PLDs were therefore rarely implanted. If necessary, a large PLD with the waist design was implanted first. In the event of residual leakage, a twist-design PLD was then added.

For the other devices, the experts usually oversized by 2 mm or 50%.

Other factors that affected device selection were catheter characteristics and the largest delivery-sheath size that will fit through the defect. For example, if only a 6-Fr sheath crosses the defect, the AVP 3 14 × 5 mm was the largest device that could be used. The largest device for a 5-Fr sheath was the AVP 3 10x5 mm and for a 4-Fr diagnostic catheter the AVP4 8 mm.

### 3.7. Specific Situations

#### 3.7.1. Large Paravalvular Leaks (PVLs)

When targeting a large defect, careful selection of the material, approach, and strategy during the preprocedural planning was crucial. Predicting whether multiple devices were required was also essential. Implantation of a single device was preferred whenever feasible. Several strategies were reported for multiple-device implantation during a single procedure (Table 11).

##### Parallel Advancement of Multiple Sheaths and Devices

The strength of this strategy is that all devices were positioned, and their efficacy assessed, before they were released. To insert parallel wires through the leak, multiple vascular approaches can be used, or multiple guidewires can be placed inside a very large, steerable sheath. For large mitral PVLs, the double transapical approach was sometimes considered.


*Tips and tricks*


To minimize the number of vascular accesses, a very useful approach is to insert a large (up to 26 French) Gore Dryseal sheath (Gore, Flagstaff, AZ, USA) in the femoral vein for mitral anterograde approach. For example, Smolka and al. illustrated that three 10 × 3-mm AVP 3 devices were implanted simultaneously through three 6-Fr guiding catheters (can even go within a 14-Fr sheath).

##### Multiple Wires in the PVL Followed by Sequential Device Deployment

In this strategy, several guidewires were inserted through the leak. One was used to track a delivery sheath and deploy a device. The corresponding wire was removed and the remaining wires, termed buddy wires, were left to maintain access through the PVL. This method eliminated the need to re-cross the PVL.

##### Sequential Implantation

A third strategy consisted in implanting each device using the same technique as for a single device. This technique was mainly used when multiple distinct PVLs were targeted during the same procedure. When there was a single large PVL, in contrast, re-crossing the channel after implanting the first device could be very challenging. Moreover, mobilization or embolization of the first device sometimes occurred during the manoeuvres performed to deploy the additional devices.

#### 3.7.2. Small Paravalvular Leaks (PVLs)

Small targeted PVL were associated with haemolysis. The main challenge lied in fitting the wire then the delivery system through the channel. Channels that were tortuous, C-shaped, S-shaped, or heavily calcified were particularly difficult to cross. A 0.035-inch or, exceptionally, a 0.014-inch guidewire was used. Strong support was often needed to advance the delivery sheath, and a stiff wire was therefore generally substituted for the first guidewire. Advancing the sheath over a 0.035-inch, stiff guidewire improved sheath push ability, making the defect easier to cross. Using a sheath with its introducer also facilitated crossing. A lower device profile enabled use of a narrower sheath that was more likely to cross the PVL. In addition, the mother-in-child technique and/or support from a steerable sheath were useful. When the delivery system could not be passed through the defect, careful balloon dilation of the leak was exceptionally performed (see Table 9) When only a 0.038-inch probe could be advanced through the defect, only an AVP4 could be implanted.

### 3.8. End of the Procedure

At the end of the procedure, complete elimination of the leak was the best outcome. However, this goal was rarely achieved. A minor-to-mild residual regurgitant leak was usually tolerated but was sometimes associated with persistent or worsening haemolysis after the procedure. Moderate-to-severe residual leaks were viewed as procedural failures. With larger PVLs, the left atrial and pulmonary pressures dropped immediately after the tPVLc, while the arterial pressure increases.


*Tips and tricks*


Comparing multiple pre- and -postprocedural hemodynamic and echocardiographic parameters was usually carried out to assess PVL reduction.

## 4. Discussion

This collaborative survey of experts and review of current devices and techniques provides insights into the complexity and technical difficulties raised by tPVLc. The many practical tips and tricks learned over a decade of experience have certainly improved results and outcomes, together with improvement of the imaging and the tools including the development of two tPVLc-specific occluding devices. A significant learning curve effect for tPVLc had been previously demonstrated [35]. Low volume of procedures had also been previously related to procedural results [17]. All 12 experts emphasized the need for high-quality teamwork among interventionalists, surgeons, echocardiographers, and other imaging specialists.

Further improvements in materials for tPVLc are expected. The surveyed experts had many suggestions for addressing unmet needs. One of the main issues is residual leakage at the end of the procedure, which may result in persistent heart failure and persistent or worsening haemolysis. To determine whether a residual leak is acceptable or likely to cause haemolysis, a test that could be performed within one minute before ending the procedure would be useful. New devices should be designed with the goal of minimizing the risk of residual leakage. The ideal device would be self-expandable, provide complete sealing via a perfect match to PVL dimensions, and have no effect on the valve leaflets or other neighbouring structures. A lower profile, particularly for the PLD, and low-profile sheaths would facilitate PVL crossing and device implantation.

Larger devices are expected to be more effective in closing PVLs, particularly the labelled AVP 3 device, whose largest dimensions of disc and waist are to date 14 mm in and 5 mm. For the PLD, the experts suggested that availability of versions with an inter-disc distance greater than 6–7 mm might provide greater effectiveness, notably for tortuous mitral PVLs. Both labelled tPVLc-specific devices are made of nitinol, which has shape memory. Additional tPVLc-specific devices are awaited. A crescent-shaped device may better match the morphology of many PVLs. Adding another material inside the occluder or a skirt on the ventricular side has been suggested to decrease the risk of residual leakage. For PVLs after trans-aortic valve implantation, a device with a longer connection producing greater flexibility might be helpful. Advances in echocardiography have greatly contributed to the development of tPVLc. However, the severity and morphology of the PVL can be difficult to assess accurately. Materials characterized by greater echogenicity, such as echogenic guidewires, would facilitate echocardiographic guidance.

Experts must continue to describe their experience and the outcomes of tPVLc. The place of tPVLc in the therapeutic algorithm may change in the near future, requiring updates of current guidelines. This procedure was first offered as a compassionate option for patients with contra-indications to surgery but is now the first-line option in many centres for patients with suitable PVL morphology. Re-operation is recommended if the PVL is related to infective endocarditis, causes haemolysis requiring repeated blood transfusions, or results in severe, symptomatic heart failure (Grade I, level of evidence C). Both American College of Cardiology/American Heart Association (ACC-AHA) and European Society of Cardiology (ESC) guidelines now recommend tPVLc for high-risk or inoperable patients, except when there is an active infective endocarditis [36,37]. However, tPVLc are complex procedures requiring lot of expensive materials and currently remain not affordable in some developing countries where incidences of rheumatic valve disease and valve replacements are high.

The on-going self-funded EuroPVLc registry (ClinicalTrials.gov Identifier: NCT05506293, accessed on 18 August 2022) was designed to promote collaborative work. Any additional centres willing to participate are welcome. The feasibility of aortic and mitral tPVLc has been demonstrated and evidence of high technical-success rates published. However, more data are needed on patient outcomes, notably in the long term. In North America, the Paradigm study funded by Abbott (ClinicalTrials.gov Identifier: NCT04489823, accessed on 28 July 2020) is a prospective, multicentre, single arm study to demonstrate the safety and effectiveness of the AVP III as a treatment for clinically significant PVLs following surgical implant of a mechanical or biological heart valve implanted in the aortic or mitral position. The trial is designed to obtained FDA approval.

## 5. Limitations

We do not report data on the full spectrum of available materials, which is already very broad. Nevertheless, we discuss the main strategies and describe the technical stratagems used by experienced operators in three international registries. We selected three large observational registries conducted by a collaborative group of experts, to obtain data on the devices used most often in recent years under clinical-practice conditions. These data may not be completely similar to data of other registries and will need updating as new materials are introduced.

## 6. Conclusions

Since tPVLc can be complex and technically demanding, considerable operator experience is required. Tips and tricks learned during more than a decade of experience may have improved technical success rates. Clinical outcomes, notably in the long term, need further investigation.

## Figures and Tables

**Figure 1 jcm-12-00119-f001:**
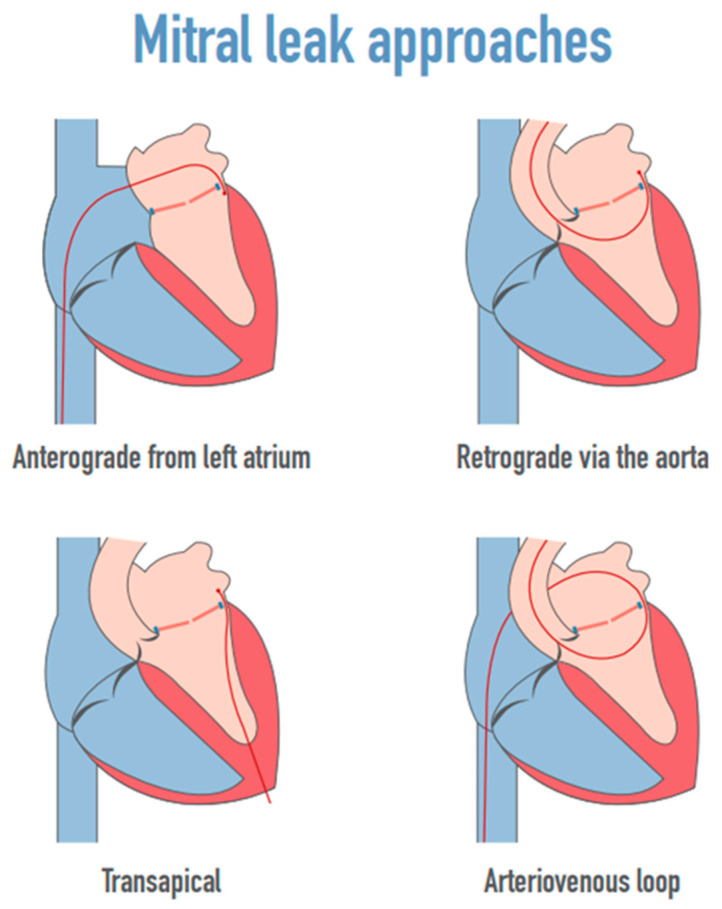
The mitral leak approaches.

**Figure 2 jcm-12-00119-f002:**
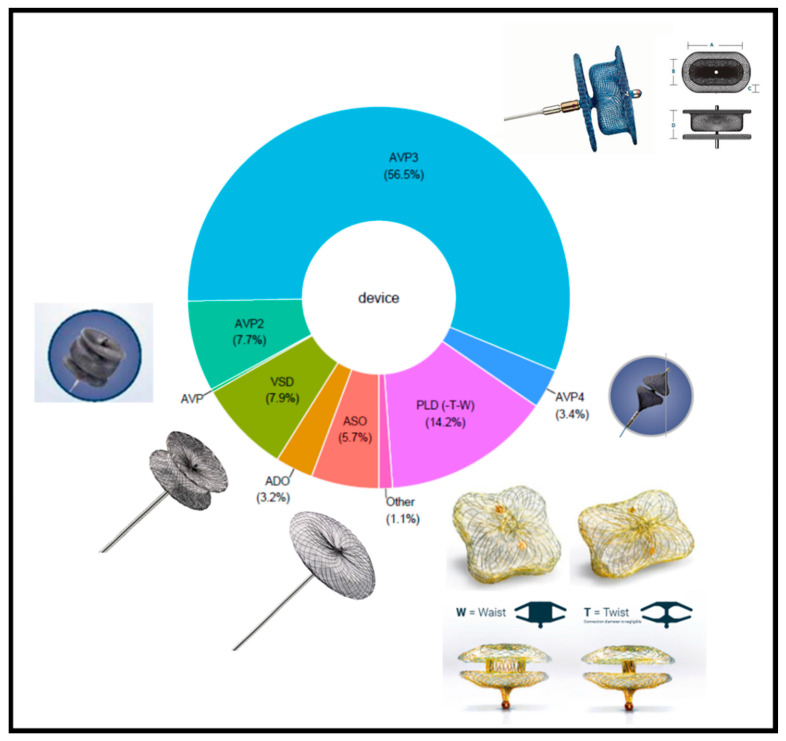
The distribution of devices in the three registries. ADO: Amplatzer Duct Occluder; ASO: Amplatzer Septal Occluder; AVP: Amplatzer Vascular Plug; AVP 3: Amplatzer Paravalvular leak plug 3; PLD: Paravalvular Leak Device (Occlutech); VSD: Ventricular Septal Defect.

**Table 1 jcm-12-00119-t001:** Imaging methods used to assess paravalvular leaks. TEE: trans-oesophageal route; CT: Computed Tomography; MRI: Magnetic Resonance Imaging.

	Pre-Procedural Imaging			
Do you use this exam before the procedure to assess PVL and plan the PVL closure approach?				
	TEE	CT scan	MRI	3D printing
expert 1	always	always	never	never
expert 2	often	sometimes	exceptionnally	never
expert 3	always	never	never	never
expert 4	always	sometimes	never	exceptionnally
expert 5	always	often	sometimes	exceptionnally
expert 6	always	sometimes	exceptionnally	exceptionnally
expert 7	always	sometimes	never	never
expert 8	always	sometimes	exceptionnally	exceptionnally
expert 9	never	sometimes	sometimes	never
expert 10	always	exceptionnally	never	never
expert 11	always	often	exceptionnally	exceptionnally
expert 12	always	always	exceptionnally	never

**Table 2 jcm-12-00119-t002:** The anaesthesia techniques for percutaneous paravalvular leak closure (tPVLc).

PVLc Guidance			
	General anesthesia	Neuroleptanalgesia	Local anesthesia
expert 1	always	never	never
expert 2	always	never	never
expert 3	always	never	never
expert 4	always	exceptionnally	never
expert 5	always	never	exceptionnally
expert 6	often	never	exceptionnally
expert 7	always for mitral/sometimes for aortic	never	often for aortic PVLc
expert 8	exceptionnally	always	always
expert 9	always	never	never
expert 10	always for mitral/sometimes for aortic	never	often for aortic PVLc
expert 11	often	never	often
expert 12	always	never	never

**Table 3 jcm-12-00119-t003:** Mitral leak approaches.

Mitral tPVLc		
	What is your first choice approach for mitral tPVLc?	Do you use steerable sheath?
expert 1	Femoral and transseptal	always
expert 2	Femoral and transseptal	always
expert 3	Transapical	never
expert 4	Femoral and transseptal	always
expert 5	Transapical	if transseptal
expert 6	Femoral and transseptal	often
expert 7	Femoral and transseptal	always
expert 8	Femoral and transseptal	always
expert 9	Femoral and transseptal	always
expert 10	Femoral and transseptal	always
expert 11	Femoral and transseptal	sometines
expert 12	Femoral and transseptal	always

**Table 4 jcm-12-00119-t004:** The aortic PVL approach.

Aortic tPVLc		
	What is your first choice approach for aortic PVLc?	Do you use alternative approach?
expert 1	femoral retrograde	no
expert 2	radial retrograde (almost 75%)	femoral retrograde
expert 3	femoral retrograde	axillary retrograde
expert 4	femoral retrograde	radial and retrograde
expert 5	femoral retrograde	left subclavian retrograde
expert 6	femoral retrograde	no
expert 7	radial retrograde	femoral retrograde
expert 8	femoral retrograde	humeral or radial retrograde
expert 9	femoral retrograde	no
expert 10	radial retrograde	femoral retrograde
expert 11	femoral retrograde	radial and retrograde
expert 12	femoral retrograde	no

**Table 5 jcm-12-00119-t005:** The list of delivery sheaths and support steerable sheaths that were used for percutaneous paravalvular leak closure (tPVLc).

Company	Product Name	Size (Fr)	Internal Diameter (inch)	Maximal Length (cm)
Delivery sheaths that will cross the defect				
Cordis	Tempo	4	0.038	125
Abbott	TorqVue LP	4; 5	0.046; 0.059	80
Abbott	TorqVue/Trevisio	6; 7; 8; 9	0.08; 0.10; 0.11; 0.12	80
Abbott	Torqvue 2	5; 6; 7	0.072; 0.083; 0.096	120
Medtronic	Launcher guide catheter	5; 6; 7; 8	0.058; 0.071; 0.081; 0.090	110
Terumo	Destination	5; 6; 7	0.074; 0.087; 0.100	90
Cook	Flexor	4; 5; 6; 7; 8	0.0595; 0.074; 0.087; 0.100; 0.113	110
Cook	Performer	9; 10	0.117; 0.130	85
Support steerable sheath				
Abbott	Agilis	8.5	0.110/Compatible with 6 Fr delivery sheath	71—usable length to the left atrium91—total lumen length with handle
Medtronic	FlexCath advance steerable sheath	15	Internal diameter of 12 FrCompatible with 10.5 Fr delivery sheathCompatible with three 6 Fr guiding cathetersCompatible with two 6 Fr introducers	65—usable length to the left atrium81—total lumen length with handle
Oscor	Destino steerable sheath	13.8	0.181 inCompatible with three 6 Fr guiding cathetersCompatible with two 6 Fr introducers	71—usable length to the left atrium89—dilator length

**Table 6 jcm-12-00119-t006:** List of devices implanted in patients included in the retrospective FFPP registry, prospective FFPP registry, and EuroPVLc registry.

Variable	Retrospective FFPP Study (2005–2019) (N = 639)	Prospective FFPP Study (2017–2019) (N = 392)	EuroPVLc (2020–2021) (N = 191)	Total (N = 1222)	*p*-Value	Test
device, n/N (%)					<0.001	(a)
- AVP3	394/639 (61.66%)	161/392 (41.07%)	136/191 (71.20%)	691/1222 (56.55%)		
- PLD (-T-W)	92/639 (14.40%)	48/392 (12.24%)	34/191 (17.80%)	174/1222 (14.24%)		
- VSD	24/639 (3.76%)	69/392 (17.60%)	3/191 (1.57%)	96/1222 (7.86%)		
- AVP2	25/639 (3.91%)	63/392 (16.07%)	6/191 (3.14%)	94/1222 (7.69%)		
- ASO	45/639 (7.04%)	24/392 (6.12%)	1/191 (0.52%)	70/1222 (5.73%)		
- AVP4	11/639 (1.72%)	21/392 (5.36%)	9/191 (4.71%)	41/1222 (3.36%)		
- ADO	39/639 (6.10%)	0/392 (0.00%)	0/191 (0.00%)	39/1222 (3.19%)		
- Other	6/639 (0.94%)	6/392 (1.53%)	2/191 (1.05%)	14/1222 (1.15%)		
- AVP	3/639 (0.47%)	0/392 (0.00%)	0/191 (0.00%)	3/1222 (0.25%)		
(a) Pearson’s Chi-square						

ADO: Amplatzer Duct Occluder; ASO: Amplatzer Septal Occluder; AVP: Amplatzer Vascular Plug; AVP 3: Amplatzer ParaValvular plug III; PLD: Paravalvular Leak Device (Occlutech); VSD: Ventricular Septal Defect.

**Table 7 jcm-12-00119-t007:** The list of Abbott devices used in the 3 registries with their compatibility with various delivery sheaths. AVP 4 were delivered through a 4-Fr braided diagnostic catheter with a 0.038-inch inner lumen.

Device Name	Guiding Catheter Medtronic Launcher	Destination	Flexor	Torqvue
AVP 2 (3–4–6–8)	5 Fr	5 Fr	4 Fr	5 Fr
AVP 2 (10–12)	6 Fr	5 Fr	5 Fr	6 Fr
AVP 2 (14–16)	NA	6 Fr	6 Fr	8 Fr
AVP 2 (18–20–22)	NA	7 Fr	7 Fr	9 Fr
AVP 3 (4 × 2–6 × 3)	6 Fr	5 Fr	5 Fr	5 Fr
AVP 3 (8 × 4–10 × 3)	6 Fr	5 Fr	5 Fr	6 Fr
AVP 3 (10 × 5)	NA	5 Fr	5 Fr	6 Fr
AVP 3 (12 × 3–12 × 5–14 × 3–14 × 5)	NA	6 Fr	6 Fr	7 Fr
ASO (4, 5, 6, 7, 8, 9, 10)	NA	6 Fr	6 Fr	6 Fr
ASO (11, 12, 13, 14, 15, 16, 17)	NA	7 Fr	7 Fr	7 Fr
VSD musc (4)	6 Fr	5 Fr	5 Fr	5 Fr
VSD musc (6, 8, 10)	NA	6 Fr	6 Fr	6 Fr
VSD musc (12)	NA	7 Fr	7 Fr	7 Fr
VSD musc (14, 16)	NA	NA	NA	8 Fr
VSD musc (18)	NA	NA	NA	9 Fr

**Table 8 jcm-12-00119-t008:** The list of PLD devices used in the 3 registries with their compatibility with various delivery sheaths.

Device Name	Guiding Catheter Medtronic Launcher	Destination	Flexor/Per–Former	Minimal Sheath Size
PLD square waist (4, 5, 6)	NA	7 Fr	7 Fr	7 Fr
PLD square waist (7)	NA	7 Fr	8 Fr	7 Fr
PLD square twist (3, 5)	NA	7 Fr	7 Fr	7 Fr
PLD square twist (7)	NA	7 Fr	8 Fr	7 Fr
PLD rectangular waist (4 × 2–6 × 3)	NA	7 Fr	7 Fr	7 Fr
PLD rectangular waist (8 × 4)	NA	7 Fr	8 Fr	7 Fr
PLD rectangular waist (10 × 4)	NA	8 Fr	8 Fr	8 Fr
PLD rectangular waist (12 × 5–14 × 6)	NA	NA	9 Fr	9 Fr
PLD rectangular waist (16 × 8–18 × 10)	NA	NA	10 Fr	10 Fr
PLD rectangular twist 5	NA	7 Fr	7 Fr	7 Fr
PLD rectangular twist 7	NA	7 Fr	8 Fr	7 Fr
PLD rectangular twist 10	NA	8 Fr	8 Fr	8 Fr
PLD rectangular twist 12	NA	NA	9 Fr	9 Fr

**Table 9 jcm-12-00119-t009:** Amplatzer Vascular Plug 4 (AVP 4) for tPVLc.

AVP 4	
	Do you consider AVP 4 if only a 0.038 inch catheter is crossing?
expert 1	never
expert 2	often
expert 3	never
expert 4	always
expert 5	never
expert 6	often
expert 7	never
expert 8	sometimes
expert 9	Sometimes—a 8mm AVP 4
expert 10	often
expert 11	exceptionnally
expert 12	sometimes

**Table 10 jcm-12-00119-t010:** The devices for tPVLc. AVP: Amplatzer Vascular Plug.

Devices For tPVLc			
	What are the devices you used preferably for tPVLc? (by order of choice)		Devices that you would not recommend to use?
	Mitral PVL	Aortic PVL	
expert 1	AVP 3	AVP 3	ASO, PFO
expert 2	AVP 3, PLD, AVP 2, AVP 4	AVP 3, PLD AVP 2, AVP 4	
expert 3	PLD	PLD	
expert 4	AVP 3, AVP 2, VSD	AVP 2, AVP 4	ASO
expert 5	PLD	PLD	ASO, VSD
expert 6	AVP 3, AVP 2, VSD, AVP 4	AVP 3, AVP 2, AVP 4	ASO, ADO2
expert 7	AVP 3, PLD, ADO 1, AVP 2, AVP 4, ADO 2	AVP 3, AVP 2, AVP 4, ADO 2, ADO 1, PLD	ASO, VSD, coil
expert 8	AVP 3, PLD	AVP 3, PLD	ASO
expert 9	AVP 2, AVP 3, ADOII, mVSD, AS0	AVP 2, ADO 2, AVP 4	
expert 10	AVP 3, AVP 4, AVP 2	AVP 3, AVP 4, AVP 2	coil
expert 11	AVP 3, PLD, ADO, AVP 2	AVP 2, ADO, AVP 3	
expert 12	AVP 3 (90%), mVSD (10%)	AVP 3 (90%), mVSD (10%)	

**Table 11 jcm-12-00119-t011:** tPVLc in large PVL.

Large PVL—Multiple Devices Strategy				
	First line approach	Technic 1. Multiple sheath and devices in parallel	Technic 2. Multiple wires in the PVL and consecutive device deployment—release	Technic 3 consecutive crossing—Implant and release of devices
expert 1	percutaneous	often	often	sometimes
expert 2	percutaneous	sometimes	sometimes	sometimes
expert 3	Transapical in Mitral PVL.Tranfemoral/axillary in aortic PVL	often	often	sometimes
expert 4	percutaneous	often	sometimes	never
expert 5	transapical	often	sometimes	sometimes
expert 6	percutaneous	sometimes	often	sometimes
expert 7	percutaneous	sometimes	often	exceptionally
expert 8	percutaneous	Often in aortic PVLprefer one large for mitral PVL	always	exceptionally
expert 9	percutaneous	often	often	often
expert 10	percutaneous	sometimes	sometimes	sometimes
expert 11	percutaneous	sometimes	sometimes	often
expert 12	percutaneous	sometimes	sometimes	often

## Data Availability

The data that support the findings of this study are available from the corresponding author, SH, upon reasonable request.

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
