# Peer review of "Procedural Tools and Technics for Transcatheter Paravalvular Leak Closure: Lessons from a Decade of Experience"

_jcm, 2022, doi:10.3390/jcm12010119_

Round 1
Reviewer 1 Report
Carefully and meticulously written, apt for an highly qualified and professional readership.The paper deals with technical aspects of PVL closure as reported by a survey by experts in the field. The approach of the manuscript is novel and timely as it is specifically addressing tips and tricks in the various clinical scenarios. The study is methodologically sound and the message is plain. The information is valuable as it derives directly from the battlefield and is entirely focused on practice patterns. Discussion is well written and references are adequate.
Author Response
Thank you for your comments
Reviewer 2 Report
Procedural Tools and Technics for Transcatheter Paravalvular 2 Leak Closure: Lessons from a Decade of Experience
The present review describes the expert views of PVL closure. It is a comprehensive overview of this complex procedure and provides tips and tricks for preprocedural imaging, each step of the procedure, and different kinds of material and devices. It reads well and is quite helpful to verify various approaches. I have few comments for the authors.
- AVP 3 or AVP 2 is preferable over AVPIII (makes it more difficult to read or differentiate from AVPII)
- What has not been addressed but represents one of the most challenging steps of this procedure, is how to best localize the leak and importantly, how the correct passage across the leak can be verified. The latter may be even more challenging in prior TAVI.
- CT may also be useful to identify the localization of the leak and the best c-arm angulation on fluoroscopy upfront. This could be immeasurable in procedures performed without TEE guidance.
- I suggest adding a figure that shows the clockwise location of mitral valve PVL and the best approach using the different access routes and the best site for transseptal puncture. This figure could either replace Figure 1 or be integrated into it.
- Table 7 is very helpful for an orientation of the suitable delivery catheter or sheath with the respective device, but I´m missing the AVP 4.
- There are few typos, for instance in Table 3 (What is your first choice pour).
- Table 9: VP4 probably means AVP 4. Please correct.
Author Response
The present review describes the expert views of PVL closure. It is a comprehensive overview of this complex procedure and provides tips and tricks for preprocedural imaging, each step of the procedure, and different kinds of material and devices. It reads well and is quite helpful to verify various approaches. I have few comments for the authors.
- AVP 3 or AVP 2 is preferable over AVPIII (makes it more difficult to read or differentiate from AVPII)
Authors’ answer: the manuscript was modified accordingly
- What has not been addressed but represents one of the most challenging steps of this procedure, is how to best localize the leak and importantly, how the correct passage across the leak can be verified. The latter may be even more challenging in prior TAVI.
Authors’ answer: Indeed, important comment, the manuscript was modified accordingly
- CT may also be useful to identify the localization of the leak and the best c-arm angulation on fluoroscopy upfront. This could be immeasurable in procedures performed without TEE guidance.
Authors’ answer: the manuscript was modified accordingly
- I suggest adding a figure that shows the clockwise location of mitral valve PVL and the best approach using the different access routes and the best site for transseptal puncture. This figure could either replace Figure 1 or be integrated into it.
Authors’ answer: We do believe that it is an interesting comment. However, we were unable to provide a figure with transseptal and mitral location in a timely manner. We apology for that.
- Table 7 is very helpful for an orientation of the suitable delivery catheter or sheath with the respective device, but I´m missing the AVP 4.
Authors’ answer: The AVP4 can be inserted through a 0.038 diagnostic probe as illustrated in the dedicated paragraph. This has been added in the legend of the figure
- There are few typos, for instance in Table 3 (What is your first choice pour).
Authors’ answer: the manuscript was modified accordingly
- Table 9: VP4 probably means AVP 4. Please correct.
Authors’ answer: the manuscript was modified accordingly
